

# Exploring private land conservation non-adopters' attendance at outreach events in the Chesapeake Bay watershed, USA

Daniel J. Read, Alexandra Carroll and Lisa A. Wainger

Chesapeake Biological Laboratory, University of Maryland Center for Environmental Science, Solomons, MD, United States of America

## ABSTRACT

**Background**. Outreach events such as trainings, demonstrations, and workshops are important opportunities for encouraging private land operators to adopt voluntary conservation practices. However, the ability to understand the effectiveness of such events at influencing conservation behavior is confounded by the likelihood that attendees are already interested in conservation and may already be adopters. Understanding characteristics of events that draw non-adopters can aid in designing events and messaging that are better able to reach beyond those already interested in conservation.
**Methods**. For this study, we interviewed 101 operators of private agricultural lands in Maryland, USA, and used descriptive statistics and qualitative comparative analysis to investigate differences between the kinds of outreach events that adopters and non-adopters attended.
**Results**. Our results suggested that non-adopters, as compared to adopters, attended events that provided production-relevant information and were logistically easy to attend. Further, non-adopters were more selective when reading advertisements, generally preferring simplicity. Future research and outreach can build on these findings by experimentally testing the effectiveness of messages that are simple and relevant to farmers' production priorities.

# INTRODUCTION

Outreach events are important venues for educating communities about, and encouraging enrollment in, conservation practices and programs (*Hall & Fleishman, 2010*). Such events are especially important for advancing conservation on private working lands, where many programs rely on agricultural producers voluntarily adopting practices that may add costs to their operations (*Lichtenberg, 2004*; *Kamal, Grodzińska-Jurczak & Brown, 2015*; *Capano et al., 2019*; *Sketch, Dayer & Metcalf, 2020*). In such situations, conservation practitioners routinely host trainings, demonstrations, field visits, and workshops to provide producers with information about how incorporating conservation practices will affect their operation

Corresponding author
Daniel J. Read, dread@umces.edu

(*Miller, Mariola & Hansen, 2008*; *Genskow, 2012*; *Christianson et al., 2014*; *Starr et al., 2015*; *Zeweld, Huylenbroeck & Speelman, 2017*).

Implementing conservation practices on private working lands can produce a number of environmental benefits (*Swinton et al., 2007*; *Kremen & Merenlender, 2018*). These diverse practices, ranging from stream restoration to minimal tillage, can reduce soil erosion and nutrient and sediment runoff, enhance wildlife habitat, and sequester carbon (*González-Sánchez, 2012*; *García et al., 2016*; *Lee et al., 2020*), though with varying effectiveness (*Osmond et al., 2012*). Individuals' reasons for whether to adopt these practices differ across several indicators, including their perceptions of how conservation practices will affect yields, their available capital and time to invest in the practice, and their environmental attitudes and levels of education (*Liu, Bruins & Heberling, 2018*; *Dessart, Barreiro-Hurlé & Van Bavel, 2019*; *Prokopy et al., 2019*; *Ranjan et al., 2019*). In response to these diverse motivations and barriers, practitioners and policy-makers use numerous techniques, including financial incentives, social marketing campaigns, and outreach events, to encourage individuals to adopt conservation practices (*Piñeiro et al., 2020*).

Recent field experiments have tested how different attributes of outreach events contribute to their effectiveness at promoting the voluntary adoption of conservation practices. Varying the distance of outreach events from participants' locale can expand attendees' advice-sharing networks and increase likelihood of adopting conservation practices (*Matous & Todo, 2018*). Adding public commitment-making and other microinterventions to outreach events has also been shown to motivate some attendees to coordinate with and recruit more neighbors to participate in conservation programs on private lands (*Niemiec et al., 2019*). However, assessments of the effectiveness of outreach events are confounded by the potential for sampling bias, in that those attending events are likely to already be interested in conservation (*Singh et al., 2018*). Thus, to ensure that such events are reaching new audiences, there is a need to understand what motivates the attendance of individuals who are less likely to adopt voluntary conservation practices.

Conservation messaging research has explored similar questions about how to motivate people's participation in conservation (*Kidd et al., 2019*). Yet results from such studies may not be directly applicable to the intermediate step of increasing attendance by non-adopters at outreach events. Here, we define non-adopters as those individuals who have not adopted any of a set of conservation practices for which they are eligible (see below). Many messaging studies test interventions drawn from behavioral economics and nudge theory (*Thaler & Sunstein, 2008*; *Byerly et al., 2018*), such as positive *versus* negative framing (*Jacobson et al., 2019*), information about social norms (*Byerly et al., 2019*), and appeals to empathy (*Czap NV, Banerjee & Burbach, 2019*), among others. However, among those studies that measure behavioral outcomes, the dependent variable is often whether respondents request further information (*e.g.*, *Dean, Fielding & Wilson, 2019*; *Reddy et al., 2020*), which is a sufficiently different behavior from attending an outreach event to warrant caution in extrapolating findings. Further, the effectiveness of conservation messaging varies greatly, with no type of messaging intervention showing a consistent and significant direction of effect.

Another limitation for applying conservation messaging research to reach non-adopters is that few studies have taken into account the production-orientation of many producers

who operate private agricultural lands and are primarily motivated by increases in yields and income (but see *Andrews et al., 2013*; *Reddy et al., 2020*). In contrast to environmentally-oriented producers, production-oriented producers have few non-economic motives for adopting conservation. Such individuals are less likely to adopt practices that take land out of production, but may be interested in practices with short-term financial gains (*Moon & Cocklin, 2011*; *Guillem et al., 2012*; *Daloğlu et al., 2014*; *Daxini et al., 2019*; *Upadhaya, Arbuckle & Schulte, 2021*). While trust, identity, and other factors do influence adoption decisions, time-management, profits, and yields are overriding concerns, particularly among US agricultural producers (*Osmond et al., 2012*).

As a first step towards understanding what messages might motivate people with different propensities for conservation to attend outreach events, we conducted 101 phone interviews to learn how farmers in Maryland, USA, respond to advertisements for outreach events and what kind of outreach events they attend. In doing so, we aimed to address the question: How is the attendance of conservation non-adopters at outreach events influenced by characteristics of the producer, messaging, and the event? Our results suggested that non-adopters were much more selective in what advertisements they decided to read, and that they decided which events to attend largely based on the practicality of attending, in terms of logistics and whether they thought the information they would learn at the event would offset time and other costs.

## MATERIALS & METHODS

### Study site

We conducted interviews with crop and livestock producers in Maryland, USA. Maryland farm production is diverse but, similar to much US cropland, is dominated by corn, soybean, wheat, and barley. Chickens are the dominant animals produced and the industry has undergone substantial consolidation over the past 20 years (*USDA, 2017*; *DCA, 2021*). Maryland differs from other regions by having a relatively small average farm size of 160 acres compared to the national average of 441 acres. Agricultural lands in Maryland face pressure from urbanization. Land cover data developed by USGS suggests a median loss of 9% of farmland in Maryland counties between 1983–2013, with a few counties experiencing losses of about 11,000–13,000 acres over this period (*Irani & Claggett, 2017*). Similarly, USDA data suggest that between 1997–2017, the number of farms in Maryland declined by about 6% (*USDA, 2017*).

Rates of adoption for agricultural conservation practices, particularly cover crops, are relatively high in Maryland (*Wallander et al., 2021*). This high adoption rate is largely due to decades of effort to reduce nutrient runoff to the Chesapeake Bay, which is a eutrophic estuary and the receiving water body for most of Maryland's agricultural land. To reduce agricultural nutrient runoff, Maryland has passed regulations and expanded cost-share programs (*Fleming, 2017*), including substantially higher payments for some cover crops in comparison to other states in the watershed (*Bowman & Lynch, 2019*). Additionally, a diverse array of county, state, and federal agencies, and non-profit organizations, host events that include one-hour webinars, farm tours, and annual conferences or workshops.

These events complement other outreach initiatives, such as one-on-one farm visits, mentorship programs, and mass media communications, which are aimed at encouraging the adoption of voluntary agricultural conservation practices.

Improving the effectiveness of outreach efforts at engaging new audiences is crucial to achieving further conservation practice implementation and water quality goals. Water quality monitoring suggests that agriculture remains the largest contributor of nonpoint source nutrient runoff into the Chesapeake Bay and that such runoff has not substantially declined over the past thirty years (*Ator et al., 2020*). This lack of substantial decline in agricultural nutrient runoff makes understanding how to reach non-adopters particularly important to future restoration initiatives in the Chesapeake Bay watershed.

**Data collection**

We conducted phone interviews between September-November 2020 with farmers selected to represent diverse production types. We constructed a sampling frame using several public directories from county and regional websites that connect producers and consumers (Supplemental Information). Because these directories overrepresented vegetable farmers, we also sampled farmers using a publicly available farm subsidy database (*Environmental Working Group, 2020*). Using this database, we randomly selected two individuals per county, one who had received a corn subsidy and one who had received a soybean subsidy. We selected these individuals using a random number generator to select an initial farmer on the list, and then randomly moved up or down the list until we selected a farmer whose contact information was available online.

We used multiple email and phone contacts to collect responses. If we had an email address, we sent an initial email and followed up with a second email if we did not receive a reply after one week (Supplemental Information). We then called remaining non-respondents by phone and left a callback number on voice message if no one answered (*Dillman, Gallegos & Frey, 1976*; *Vogl et al., 2019*). If we did not have an email address, we called farmers, leaving a voice message if no one answered. If we did not receive a reply after one week, we called a second time and left another voice message. We excluded any respondent who also worked for conservation organizations like the USDA Natural Resources Conservation Service (NRCS), soil and water conservation districts (SWCD), university extension, or a non-profit conservation organization.

During the interviews, we asked respondents about their operation, their use of conservation practices, and whether they attended any outreach events in 2019. If they had, we asked about attributes of the most recent event they attended, including what it was about, who organized it, how they heard about it, and what their motivations were for attending. If they reported not attending any outreach events, we asked them if they had seen any advertised, and if so, why they chose not to attend and what might motivate them to attend events in the future (Supplemental Information). We pre-tested our questionnaire with four farmers in Virginia, the state directly south of Maryland, and made slight changes to the question order and wording before administering the interviews in Maryland. All respondents gave verbal informed consent to participate in this study and the University of

Maryland, College Park Institutional Review Board approved all data collection procedures (#1524456).

## Data analysis

With respondents' permission, we audio recorded the interviews and coded them using directed content analysis (*Hsieh & Shannon, 2005*; *Bernard & Ryan, 2010*). We created a codebook with possible responses for each question (Supplemental Information), and marked each code that the respondents mentioned as 1, and all others as 0. To assess intercoder reliability, each coder independently coded five interviews already coded by another coder, and we compared the two sets of codes by calculating Cohen's kappa (*Cohen, 1960*). The resulting kappa value was 0.838, indicating that the agreement between the coders was 83.8% better than expected by chance.

We analyzed the data and distinguished non-adopters from adopters using both descriptive statistics and qualitative comparative analysis (QCA; *Basurto, 2013*; *Pahl-wostl & Knieper, 2014*; *Brockhaus et al., 2017*). QCA applies techniques from Boolean algebra and set theory to identify key combinations of conditions associated with an outcome (*Ragin, 1987*; *Rihoux, 2003*; *Grofman & Schneider, 2009*). It is particularly suitable for analyzing moderately-sized datasets that are too small for standard statistical techniques but too large for in-depth qualitative case study analysis (*Ragin et al., 2003*).

The codes from the content analysis formed the basis of the data calibration for QCA (note that QCA uses the terms 'conditions' and 'calibration' instead of 'variables' and 'coding,' respectively). We calibrated respondents as eligible for different conservation practices based on what they produced. If they produced livestock, we classified them as eligible for three livestock-related practices, and if they produced crops, we classified them as eligible for four crop-related practices (Table 1). We chose these practices because they are currently heavily promoted by policy-makers and staff from government and non-profit organizations in Maryland. Further, they apply to individual farming operations regardless of the physical landscape, allowing us to determine eligibility based on farm production characteristics alone.

Using these calibrations for eligibility, we then created two main outcome conditions that distinguished whether respondents were adopters or non-adopters and whether they reported attending at least one outreach event. Non-adopter attendees were those respondents who did not report adopting any of the conservation practices for which they were eligible and reported attending at least one outreach event. Adopter attendees were those who reported adopting at least one of the conservation practices for which they were eligible and reported attending at least one outreach event. We chose this restrictive definition of non-adoption, rather than considering practices individually, because we were particularly interested in understanding what kinds of events are attended by individuals who rarely interact with conservation practitioners. Adopting one conservation practice significantly increases the likelihood of adopting other complementary practices (*Fleming, 2017*; *Prokopy et al., 2019*; *Canales, Bergtold & Williams, 2020*), thus effectively creating a subset of farmers who routinely interact with conservation practitioners. By classifying non-adopters as those who have not adopted any practices for which they are eligible, we

**Table 1  Description of conservation practices.**

| Production type | Conservation practice | Description | Citation |
|---|---|---|---|
| Livestock | Stream exclusion fencing | Livestock are prevented from entering streams by fences to reduce stream bank erosion and deposition of animal waste in the water | *Bragina et al. (2017)* |
| | Rotational grazing/pasture management | Livestock are frequently moved between paddocks to prevent them from using other parts of the pasture while biomass regenerates | *Sovell et al. (2000)* |
| | Manure storage facility | Manure is stored securely in concrete structures to prevent waste runoff, especially during rainy weather | *Meals & Braun (2006)* |
| Crops | Cover crops | Crops planted either to cover soil between rows or across fields during the off-season to retain soil nutrients and prevent surface runoff | *Dabney, Delgado & Reeves (2001)* |
| | Conservation tillage | Various techniques to reduce soil disturbance during planting and harvesting to prevent erosion | *Holland (2004)* |
| | Crop rotation | Plots are planted with different crops across growing seasons to increase soil microbial diversity and control pests | *Venter, Jacobs & Hawkins (2016)* |
| | Variable rate application | The rate of seed, pesticide, and fertilizer application is altered depending on specific attributes of the field, potentially reducing overall nutrient inputs | *Fleming et al. (2000)* |

aimed to distinguish those who rarely interact with conservation professionals from those who are likely to have more frequent interactions.

We calibrated 11 conditions related to characteristics of the respondent and their farm operation and 9 conditions related to characteristics of the most recent outreach event they attended (Table 2). We calibrated most conditions for the outreach events as multi-value by grouping together codes from the content analysis (*Thiem, 2015*). We calibrated conditions related to the respondents and their farm operation largely as binary, retaining the original content analysis codes. However, we calibrated total acres as a multi-value condition, using quartiles as thresholds between values. We analyzed livestock and crop producers separately so that conditions about livestock did not apply to crop producers and vice versa. Accordingly, we used 7 conditions related to the respondent and their operation in each analysis. We transformed our calibrated data into truth tables and performed a logical minimization using the consistency cubes method (*Dușa, 2018*) to produce the most parsimonious solution (*Baumgartner & Thiem, 2020*). A truth table is a reorganization of calibrated data, such that the columns represent the different conditions, each cell gets values from the calibrated data (0/1 for binary conditions, 0, 1, 2, …for multi-value conditions), and there are as many rows as there are observed combinations of those conditions. Truth tables also include whether that row led to the outcome, and the number

**Table 2  The conditions and their calibration as used in the study.**

| | | Condition | Calibration | Description |
|---|---|---|---|---|
| Remote conditions (farm characteristics) | For livestock producers | CT | Binary (1/0) | Does the farmer produce beef or dairy cattle? |
| | | SR | Binary | Does the farmer produce small ruminants like goats, sheep, or pigs? |
| | | OL | Binary | Does the farmer produce other livestock that are not cattle or small ruminants (e.g., horses, llama, alpaca, emu)? |
| | | ACl | Multi-value | For livestock producers, how many acres do they operate (quartiles)? (0 = 1–50; 1 = 51–150; 2 = 151–321; 3 = 322+) |
| | For crop producers | VG | Binary | Does the farmer produce vegetables? |
| | | GR | Binary | Does the farmer produce grains like corn, soybeans, or wheat? |
| | | OC | Binary | Does the farmer produce crops other than vegetables or grain (e.g., herbs, fruits, etc.)? |
| | | ACc | Multi-value | Multi-value, for crop farmers, how many acres do they operate (quartiles)? (0 = 0–25; 1 = 26–72; 2 = 73–299; 3 = 300+) |
| | For all producers | BF | Binary | Are they a beginning farmer according to the NRCS definition (farming fewer than 10 years)? |
| | | TE | Multi-value | What is the farmer's tenure status regarding the land? (0 = Non-operating owner; 1 = Operating owner; 2 = Non-owner operator; 3 = Own some, rent some) |
| | | AT | Multi-value | In 2019, how frequently did the farmer attend outreach events? (0 = None or one; 1 = once every few months; 2 = once a month or more often) |

Read et al. (2021), *PeerJ*, DOI 10.7717/peerj.11959

**Table 2** (*continued*)

| | Condition | Calibration | Description |
|---|---|---|---|
| | EA | Multi-value | What was the most recent event the farmer attended about? (0 = Don't remember; 1 = Other; 2 = Marketing & business; 3 = Agricultural land management; 4 = Conservation; 5 = Multiple) |
| | CH | Multi-value | How did the farmer learn about the event? (0 = Don't remember; 1 = Helped organize; 2 = Electronic, email or social media; 3 = Paper, mail or flyer; 4 = Multiple[a]) |
| | EO | Multi-value | Who organized the most recent event the farmer attended? (0 = Don't remember/no data; 1 = NRCS or SWCD; 2 = County or university extension; 3 = Non-profit; 4 = Other, private, local farmer, etc.) |
| Proximate conditions (outreach event characteristics) | MT | Multi-value | What motivated the farmer to attend? (0 = No data; 1 = Multiple; 2 = Incentives, certification credits, food, etc.; 3 = Logistics[a], online, nearby, low cost, duration, etc.; 4 = Social, knew people going or organizers, etc.; 5 = Content, topic, guest speaker, being involved in organizing the event, etc.) |
| | ED | Multi-value | How long was the most recent event the farmer attended? (0 = Don't remember; 1 = 1 − 2 hours; 2 = 2 − 4 hours; 3 = All day; 4 = Multiple days) |
| | KO | Multi-value | For the most recent event the farmer attended, did they know others going beforehand? (0 = Don't remember; 1 = No; 2 = Yes) |
| | ET | Multi-value | When during the day did the most recent event the farmer attended begin? (0 = Don't remember; 1 = Morning; 2 = Afternoon/evening) |
| | EC | Multi-value | Was there a cost associated with the most recent event the farmer attended? (0 = Don't remember; 1 = No; 2 = Yes) |
| | WK | Multi-value | Was the most recent event the farmer attended on the weekday or weekend? (0 = Don't remember; 1 = No; 2 = Yes) |

**Notes.**
[a]These values did not appear in the data.

of observed cases corresponding to that row. The minimization of the truth table adheres to the following rule: if two combinations of conditions differ in only one condition yet produce the same outcome, then that condition can be considered irrelevant and removed to create a more simple expression (*Ragin, 1987*:93). For example, if a non-adopter attended an evening event that was free, and another non-adopter attended an evening event that cost money, we would consider time of day to be a relevant condition, but cost an irrelevant condition, to describe the kinds of events non-adopters attend. The minimization process proceeds iteratively until the expression cannot be made more simple.

Because of the large number of conditions, we followed the two-step QCA procedure (*Schneider, 2019*). The two-step procedure recognizes a difference between remote and proximate conditions, which are implicit in many social science analyses. Remote conditions are relatively stable over time and largely outside the reach of conscious influence. Proximate conditions vary over time and are subject to changes introduced by actors (*Schneider & Wagemann, 2006*). The two-step procedure begins by identifying necessary remote conditions that represent enabling contexts, and then includes those with the proximate conditions when minimizing the truth table (*Schneider, 2019*).

In our analysis, we analyzed characteristics of the farmer and production as remote conditions and characteristics of the outreach events as proximate conditions. First, we identified remote conditions with consistency of necessity values above 0.9 and for which no deviant cases exist (cases including the condition but also the absence of the outcome). If no conditions meet that threshold, we assumed that remote conditions are not necessary for determining the outcome and proceeded with the parsimonious logical minimization of the proximate conditions. We evaluated our results using consistency and coverage of sufficiency scores. The former indicates the proportion of cases in which both the condition and outcome are present out of all instances of the condition, and the latter describes the same, but out of all instances of the outcomes (*Ragin, 2006*). We conducted our analysis using the 'QCA' package (*Dușa, 2019*) for R statistical software (*R Core Team, 2020*).

## RESULTS

### Descriptive Statistics

We contacted 477 farmers and collected 101 valid responses for a response rate of 21.2%. Eighty-one (80.2%) respondents were eligible for crop-related conservation practices and 52 (51.5%) were eligible for livestock-related conservation practices. Among the crop producers, 63 (77.8%) reported attending at least one outreach event in 2019 and 5 of those individuals had not adopted any of the relevant crop-related conservation practices. Among livestock producers, 39 (75%) reported attending at least one outreach event in 2019 and 4 of those individuals had not adopted any of the relevant livestock-related conservation practices. 23 (22.8%) respondents reported not attending any outreach events in 2019.

These respondents were distributed across 19 of Maryland's 24 counties, and were more frequently concentrated in the north-central part of the state and less frequently in the lower Eastern Shore (Fig. S1). The distribution of farm sizes within our sample

was similar to that of Maryland overall, though with a slightly smaller proportion of small farms and higher proportion of vegetable growers (Table S1). Respondents most frequently reported producing 'Other crops,' which included mushrooms, emu, horses, fruits, and alpacas, followed by corn, hay, and beef cattle (Fig. 1). Most respondents reported attending outreach events once every few months in 2019. Among those eligible for crop-related conservation practices, nearly 80% ($N = 64$) reported using cover crops and nearly 70% ($N = 54$) reported using crop rotations. Among those eligible for livestock-related conservation practices, nearly 70% ($N = 37$) reported using rotational grazing and over 50% ($N = 28$) reported using stream exclusion fencing. Among both groups, the proportion of respondents reporting non-adoption of any eligible conservation practices was low.

Adopters and non-adopters reported important differences in how they filtered through event advertisements and how they considered which events to attend (Fig. 2). Non-adopters were much more likely than adopters to report filtering advertisements based on whether the event was about what they produce on their farm and whether it was practical to attend the event. Practicality, in this sense, refers to whether the perceived cost in time spent away from the farm to attend the event would be compensated by the usefulness of the information they might learn by attending. For example, one non-adopter explained that she considers which events to attend based on which will give "the most bang for my buck or time." Adopters were much more likely than non-adopters to report not filtering the advertisements they received and looking at everything that came through their email inbox.

Similar to the responses of advertisement filtering, non-adopters who did not attend events cited practicality and applicability of event topics as reasons for not attending, while adopters had more wide ranging responses. Of the 101 respondents, only 5 had not adopted any conservation practices for which they were eligible and did not report attending any outreach events. They reported that the main reasons they did not attend events were the lack of applicable or interesting topics and the inconvenient timing of outreach events. All these respondents said that they would be more motivated to attend events that were nearby, short, and low to no cost to attend. Nineteen respondents had adopted at least one conservation practice for which they were eligible but did not report attending any outreach events. Of these, 9 said that they did not remember receiving, or have not received, advertisements for events. Others cited lack of internet access, inconvenient times, and event information being too simple as reasons for not attending. Ten non-attendee adopters said that they would or might attend future events, and suggested that more advanced event topics, guest speakers, online events online, and more convenient timing for in-person events would motivate them to attend.

## Qualitative comparative analysis

We only included the proximate conditions (outreach event characteristics) in the logical minimization of the interview data because no remote conditions (farm characteristics) met the necessary inclusion criteria. Across all qualitative comparative analyses, no combination of remote conditions was both (1) associated with at least 90% of all instances of the

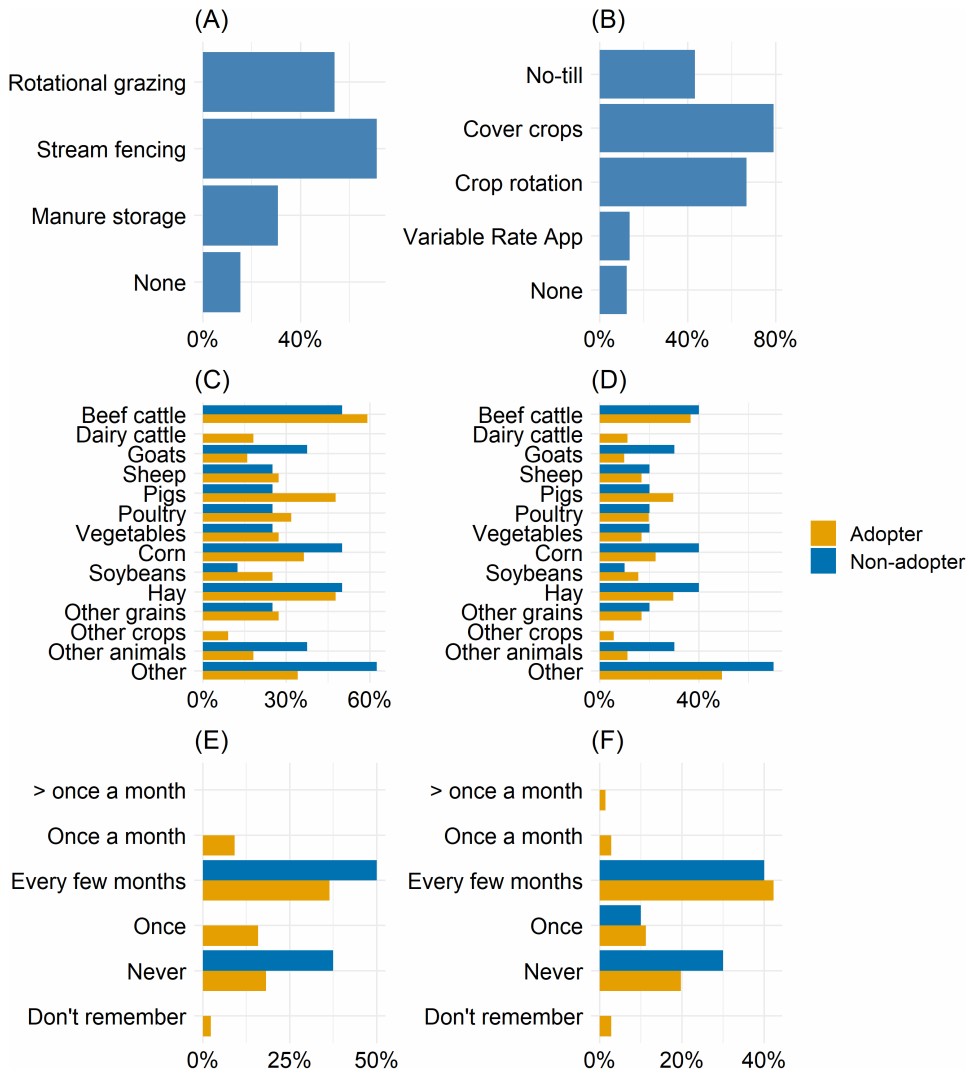

**Figure 1   Bar charts showing variation among interview respondents.** (A, B) show rates of conservation practice adoption among livestock producers (A; $N = 52$) and crop producers (B; $N = 81$). (C, D) show farm products produced by livestock producers (C) and crop producers (D). (E, F) show frequency of attended outreach events in 2019 for livestock producers (E) and crop producers (F).

outcome, and (2) not associated with any instances of the absence of the outcome. This finding held across all outcomes tested, indicating that remote conditions are not necessary for explaining whether an attendee was an adopter or non-adopter.

We identified three pathways associated with event attendees who had not adopted any livestock-related conservation practices, and six pathways for attendees who had adopted at least one such practice. Both the non-adopter and the adopter solutions fit the data well, with the consistencies both equal to 1.00 and the coverage equal to 1.00 and 0.914, respectively (Fig. 3).

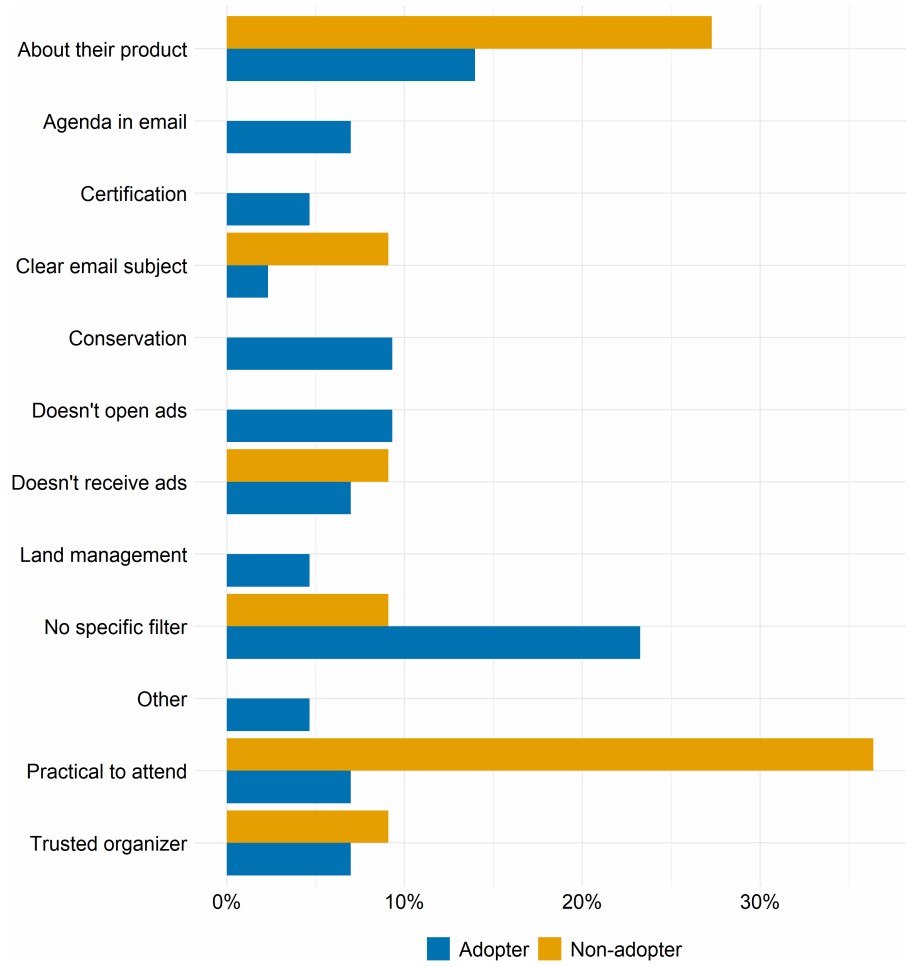

**Figure 2** Differences in how adopters and non-adopters filter advertisements they receive for outreach events.

The solution for non-adopters of livestock-related conservation practices who reported attending at least one outreach event is:

$$NA_{livestock} \Leftrightarrow ED[2]*WK[2] + MT[1]*EO[2]*ED[3] + MT[5]*EO[1]*ED[3]$$

This solution should be read as: a non-adopter of livestock-based conservation practices will have reported attending an event if: (1) the event lasted 2–4 h and occurred on the weekend, (2) they had multiple motivations for attending, the event was organized by county or university extension, and was all day, or (3) they were motivated by the event topic, the organizer was NRCS or SWCD, and it was all day.

These individuals were characterized by attending events that they deemed to be worth the time away from their farms. The individual associated with the first pathway in Fig. 3 said that her focus was on maintaining and improving practices, rather than adding more. She reported having issues with invasive weeds and said that any time she spends off-farm needs to be devoted to solving such issues. Similarly, the other three non-adopters

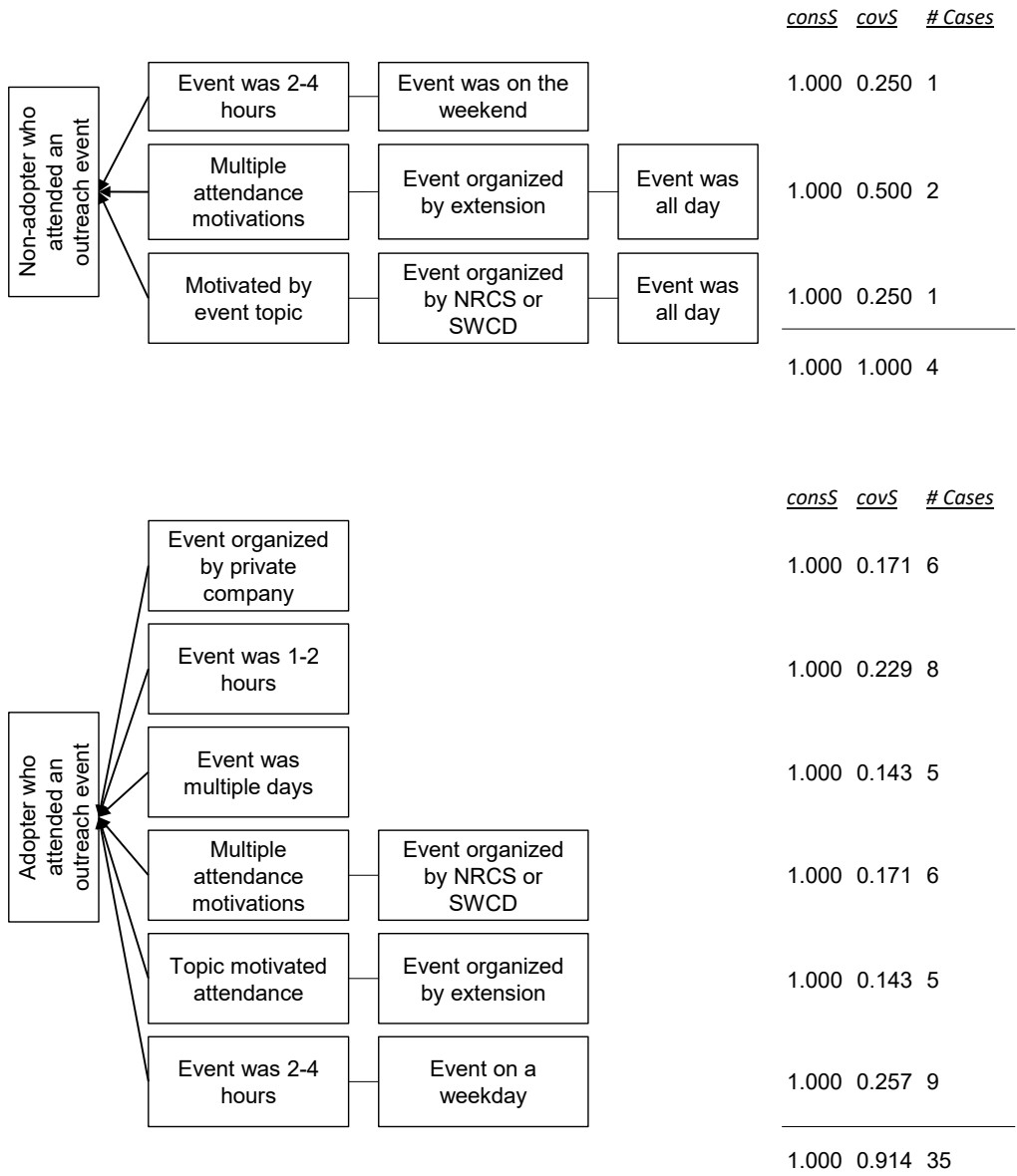

**Figure 3** **Pathways for non-adopters (top) and adopters (bottom) of livestock-related conservation practices who reported attending at least one outreach event.** Each pathway (rows) represents one combination of conditions. That is, the two boxes in the first row indicate that if someone reported attending an event lasting 2–4 h and that was on a weekend, then that respondent will be a non-adopter of livestock-related conservation practices. The consistency (consS) and coverage (covS) of sufficiency scores for each pathway, as well as the number of cases (respondents) whom this pathway describes, are shown in columns on the right. The scores for the full solution are presented at the bottom.

all reported attending events that included hands-on training and demonstrations of particular practices by other farmers. One non-adopter said that she attends events that offer "a lot of practical advice that I [can] put to use right away." The individual associated with the third pathway in Fig. 3 reported attending a farm tour that showcased a variety of livestock-related conservation practices. He emphasized that being able to see the practices
implemented, and talk with those who use them was worth the time away from his own farm.

The solution for adopters of livestock-related conservation practices who reported attending at least one outreach event is:

$$A_{livestock} \Leftrightarrow EO[4] + ED[1] + ED[4] + MT[1]*EO[1] + MT[5]*EO[2] + ED[2]*WK[1]$$

This solution should be read as: an adopter of livestock-based conservation practices will have reported attending an event if: (1) it was organized by a private company or other group, (2) the event lasted 1-2 h, (3) it was a multi-day event, (4) they had multiple motivations for attending and the event was organized by NRCS or SWCD, (5) they were motivated by the topic to attend and it was organized by a county or university extension, or (6) the event lasted 2-4 h and occurred on a weekday.

We identified four pathways associated with event attendees who had not adopted any crop-related conservation practices and ten pathways for attendees who had adopted at least one such practice. Both the non-adopter and the adopter solutions fit the data reasonably well, with the consistencies both equal to one and the coverage equal to 0.714 and 0.875, respectively (Fig. 4)

The solution for non-adopters of crop-related conservation practices who reported attending at least one outreach event is:

$$NA_{crop} \Leftrightarrow EO[0]*CH[3] + EA[5]*ET[2] + EO[0]*EC[2] + EO[4]*EC[2]*CH[2]$$

This solution should be read as: a non-adopter of crop-related conservation practices will have reported attending an event if: (1) they don't remember the organizer and received a paper advertisement about the event, (2) the most recent event they attended was about multiple topics and it started in the afternoon or evening, (3) they don't remember who organized the most recent event they attended and it cost money to attend, or (4) the event was organized by a private group, cost money, and was advertised online.

Similar to the non-adopters of livestock-related practices, these individuals were characterized by attending events that directly related to their farm production and were easy to attend. For example, the respondent associated with the third pathway in Fig. 4 reported hesitating about whether to attend because he worried it would put him behind in terms of his production goals. He attended because the cost covered a meal and because he would have the opportunity to network with other farmers like him. He said that he prefers events on other local farms, where "you can, in-person, meet a couple people and not have to drive very far." Similarly, a respondent associated with the fourth pathway in Fig. 4 reported attending an event to network and to stay familiar with industry trends. For him, the cost of attending a large trade show that enabled such networking was more than offset by the benefits it would yield to his production capabilities. The other respondent associated with the fourth pathway attended an event to stay up-to-date with manure regulations. All these respondents reported perceiving benefits of attending events that directly related to their farm production, and not to conservation generally.

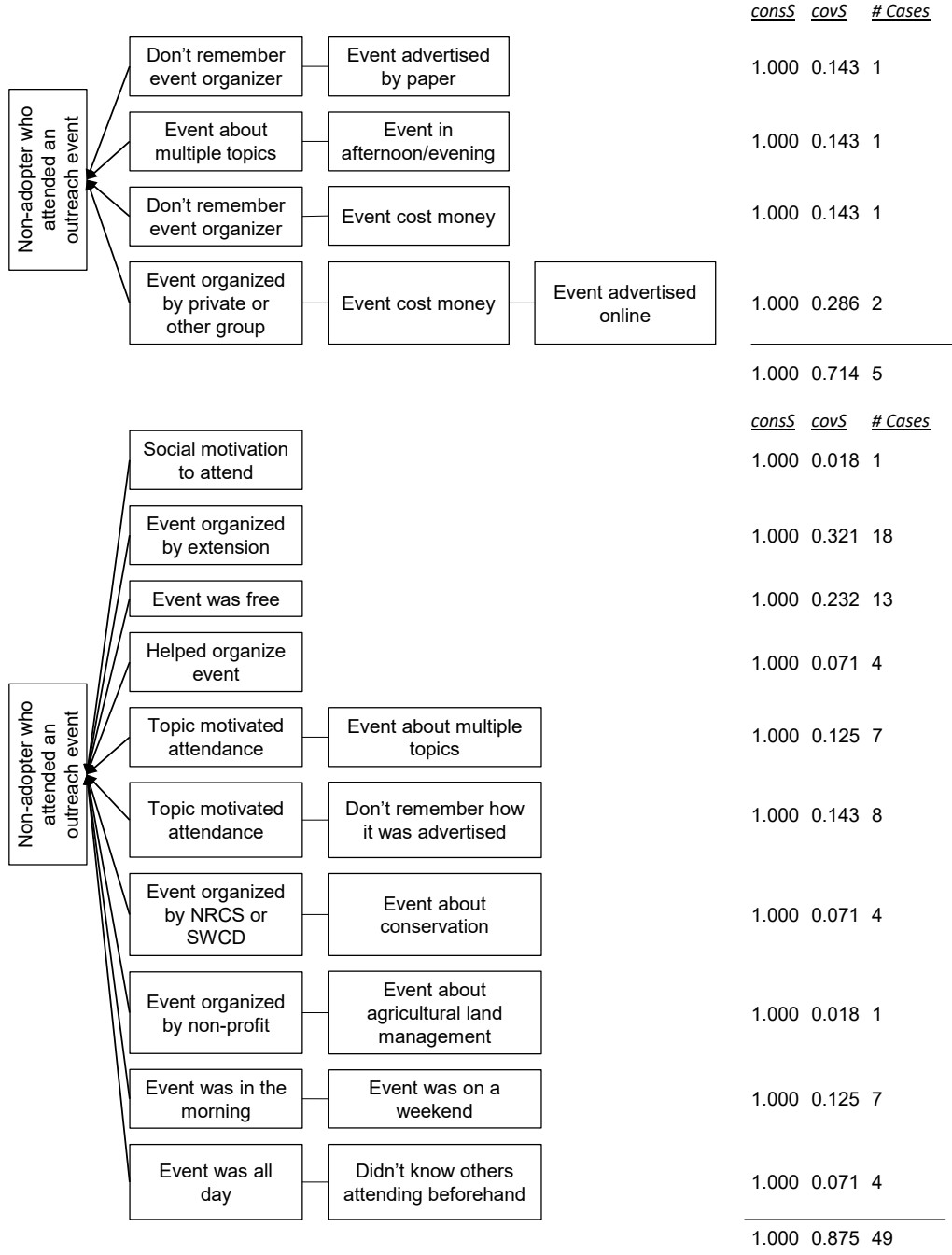

**Figure 4** **Pathways for non-adopters (top) and adopters (bottom) of crop-related conservation practices who reported attending at least one outreach event.** See Fig. 3 caption for interpretation.

The solution for adopters of crop-related conservation practices who reported attending at least one outreach event is:

$$A_{crop} \Leftrightarrow MT[4] + EO[2] + EC[1] + CH[1] + MT[5]*EA[5] + MT[5]*CH[0] + EO[1]*EA[4] + EO[3]*EA[3] + ET[1]*WK[2] + ED[3]*KO[1]$$

This solution should be read as: an adopter of crop-related conservation practices will have reported attending an event if: (1) they were motivated by a social aspect of the event, (2) the event was organized by county or university extension, (3) the event was free, (4) they helped organize the event, (5) they were motivated by the event topic and the event was about multiple topics, (6) they were motivated by the event topic and they don't remember how the event was advertised, (7) the event was organized by NRCS or a soil and water conservation district and the event was about conservation, (8) the event was organized by a non-profit and it was about agricultural land management, (9) the event was in the morning and on a weekend, or (10), the event was all day and they did not know other attending beforehand.

## DISCUSSION

Our results suggested that non-adopters, more than adopters, based their attendance decisions on their perceptions of the ease of attending and whether the value of information learned will compensate for time taken away from the farm and out-of-pocket costs (cf. *Wang et al., 2020*). Three lines of evidence supported this finding. First, non-adopters more frequently reported filtering through event advertisements for those that appeared to be convenient to attend and about something they produced, whereas adopters more frequently reported no specific filter. This emphasis on practicality is consistent with previous research on producers' engagements with conservation (*Greiner & Gregg, 2011*; *Jackson-Smith & McEvoy, 2011*), and suggests that non-adopters' decisions to attend outreach events reflected a production-orientation. Second, among those who reported not attending any events, non-adopters' reasons for not attending were largely that the event topics were inapplicable or that the events were at inconvenient times and locations.

Third, the results from the QCA further supported our finding of a production-orientation among non-adopters. One finding from the QCA suggested that non-adopters were distinguished from adopters by preferring all-day events and weekend events. For livestock-related conservation practices, adopters attended 2–4 h events on the weekday, while non-adopters attended 2–4 h events on the weekend (Fig. 3). Similarly, two non-adopter pathways including the condition that the event was all-day, while this condition was not in any adopter pathway. Respondents related the conditions of being all-day and on the weekend to how much production-relevant information they thought they would get out of the event. One non-adopter reported that he most recently attended an all-day farm tour that demonstrated several livestock-related practices. For him, the ability to see multiple relevant practices implemented and to talk to those implementing them was worth his time away from the farm. Likewise, the other non-adopters of livestock-related conservation practices suggested that they were motivated to attend events that provided hands-on training and allowed them to talk to farmers currently implementing these practices. These findings suggested that non-adopters preferred attending events that are long enough to ensure that they will learn enough information to make attendance worthwhile.

The results from the QCA also suggested that non-adopters attended events that cost money, while adopters gravitated towards free events. Among crop-related conservation

practices, only adopters reported attending free events and the condition that the event cost money appeared in two non-adopter pathways. While cost poses a small barrier to attendance (respondents reported that admission was usually between \$10–35 USD for events with fees), the respondents indicated that this barrier was more than overcome by the ability to learn different types of production-relevant information. Many conferences and trade shows have entrance fees, but also allow attendees to meet with a variety of groups and individuals, enabling them to stay up-to-date on industry and regulatory trends. Farm tours tend to be all-day events in which the cost covers a meal and attendees get to see different practices implemented on others' farms and ask questions about them. In this sense, cost, as a characteristic of events attended by non-adopters, is associated with events that provided attendees the opportunity to network with other farmers and stay up-to-date on industry and regulatory trends. While some free events may provide the same information and networking opportunities, no respondents reported any free, in-person events on the weekend. As a result, cost, in our sample, may double as a proxy for non-adopters' preference for weekend events.

Apart from distinguishing non-adopters by their emphasis on attending practical, production-relevant events, our results also showed some similarities between adopters and non-adopters. First, our analysis suggested that farmer and production characteristics were not necessary to distinguish the two groups. This finding largely agrees with the agricultural conservation adoption literature, which does not show consistent significant relationships between adoption rates and most farmer and production characteristics, with the exception of farm acreage and farmer age (*Prokopy et al., 2019*). Second, all respondents had similar motivations for attending outreach events. When respondents only had a single motivation for attending, in all cases but one, they were motivated by the event topic. When respondents reported multiple motivations for attending, all non-adopters and 90% of adopters said that one of their motivations for attending was the event topic. Third, we found only minor differences in the effect of hosting organization adopters' and non-adopters' attendance across livestock and crop farms. NRCS, SWCD, extension, non-profits, and private groups all appeared in the pathways for adopters and non-adopters.

## Recommendations for future outreach

Our exploratory results have several implications for the design of outreach events and conservation messaging to improve the likelihood of non-adopter attendance. Our results suggest that centering outreach messaging around production goals, rather than conservation, may encourage more non-adopters to attend events. More specifically, our results suggest that, to attract non-adopters, email subject lines should contain information that clarifies which production types can benefit from the practices being discussed. Further, scheduling events at convenient times with ample opportunity for farmer discussion may also encourage non-adopter attendance. Non-adopters sought in-depth information and hands-on training, as provided in all-day or weekend events, and were willing to pay to attend events that included the ability to talk to other farmers and agricultural professionals about multiple topics for significant periods of time.

### Recommendations for future research

Additional conservation messaging research could expand beyond our relatively small non-random sample and test generalizability. Our finding that farm production characteristics do not meaningfully distinguish whether event attendees are non-adopters suggests that our results about messaging and event content could apply across diverse farm types and geographies. However, safety precautions aimed at preventing the spread of COVID-19 meant that most in-person outreach events were cancelled during our research period. Future research could go beyond self-reported attendance and observe in-person events to understand how non-adopters' attendance varies by different event attributes.

More fundamentally, future work could explore how simplifying messages and emphasizing production priorities affects who attends outreach events. Our finding that non-adopters largely filter event advertisements only for relevance and practicality is consistent with messaging research that has found that greater simplicity of messages, typically measured as the amount of text or number of motivators addressed, can motivate behavior change (*Farrow, Grolleau & Mzoughi, 2018*; *John & Blume, 2018*). Much conservation messaging research evaluates whether adding information or reframing how it is presented affects respondents' behavior (*e.g.*, *Chen et al., 2009*; *Byerly et al., 2019*). Some of those studies have found that messages about increasing profits to be ineffective (*Andrews et al., 2013*; *Reddy et al., 2020*), which differs somewhat from our findings. More research is needed to examine how different production-oriented framings, such as specific production types and goals, affect people's responses to conservation messaging. Lastly, further research could examine how the number of times people are exposed to conservation messaging and the number and type of events they attend correlate with the adoption of specific conservation practices, or other non-dichotomous behavioral outcomes (*Pannell & Claassen, 2020*).

## CONCLUSIONS

Conservation messaging and outreach are more effective when they are constructed with knowledge of the social context within which they will be delivered (*Byerly et al., 2019*; *Kusmanoff et al., 2020*). Our findings from exploratory research with crop and livestock producers on private working lands in Maryland, USA support this idea and suggest that non-adopters of agricultural conservation practices used different criteria than adopters for choosing which outreach events to attend. Compared to adopters, non-adopters primarily attended outreach events that they thought would justify time taken away from the farm by providing them with production-relevant information. When filtering through event advertisements, they looked for events that were relevant to their production, convenient to attend, and that allowed ample time to talk with other farmers and agricultural professionals. We suggest that further experimental research examine the effect of simple productivity messaging and event structure on non-adopters' attendance and conservation practice adoption. Such research will help to design more effective advertisements and events that reach beyond those already interested in conservation and encourage those who have had limited engagement to explore further opportunities for adopting conservation practices.

## ACKNOWLEDGEMENTS

We thank the respondents for generously offering their time and perspective. We declare no conflicts of interest.

### Funding

This work is supported by Sustainable Agricultural Systems grant no. 2019-68012-29904/project accession no. 1019799 from the USDA National Institute of Food and Agriculture, and a grant from the National Wildlife Federation (NWF ID: 2008-029). The funders had no role in study design, data collection and analysis, decision to publish, or preparation of the manuscript.

### Grant Disclosures

The following grant information was disclosed by the authors:
USDA NIFA Sustainable Agricultural Systems: 2019-68012-29904, project accession no. 101979.
National Wildlife Federation:  NWF ID: 2008-029.

### Competing Interests

The authors declare there are no competing interests.

### Author Contributions

- Daniel J. Read conceived and designed the experiments, performed the experiments, analyzed the data, prepared figures and/or tables, authored or reviewed drafts of the paper, and approved the final draft.
- Alexandra Carroll performed the experiments, analyzed the data, prepared figures and/or tables, authored or reviewed drafts of the paper, and approved the final draft.
- Lisa A. Wainger conceived and designed the experiments, analyzed the data, authored or reviewed drafts of the paper, and approved the final draft.

### Human Ethics

The following information was supplied relating to ethical approvals (i.e., approving body and any reference numbers):

All respondents gave informed consent to participate in this study and the University of Maryland, College Park Institutional Review Board approved all data collection procedures (#1524456).

### Data Availability

The raw data and replication code for R are available in the Supplementary File.

### Supplemental Information

Supplemental information for this article can be found online at http://dx.doi.org/10.7717/peerj.11959#supplemental-information.

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
