# Peer review of "Exploring private land conservation non-adopters’ attendance at outreach events in the Chesapeake Bay watershed, USA"

_PeerJ, doi:10.7717/peerj.11959_

## Round 0.1 · original submission · Minor Revisions

· Academic Editor

Minor Revisions

I have completed the review of your manuscript, and a summary is appended below.

In general, the reviewers recommend the reconsideration of your paper following a minor revision. I invite you to resubmit your manuscript after addressing all reviewers' comments.

When resubmitting your manuscript, please carefully consider all issues mentioned in the reviewers' comments, explain every change made, and provide suitable rebuttals for any remarks not addressed.

PeerJ policy states that additional references suggested during the peer-review process should only be included if the authors agree that they are relevant and useful.

I look forward to receiving your revised manuscript.

·

Basic reporting

1. Your description of non-adopters as defined in the methods section is sufficient, but it will improve understanding of the introduction if you provide a short description of non-adopters in the introduction as well.

2. The section describing adoption of conservation practices in the study site in lines 118-129 could benefit from a discussion of conservation outreach/implementation initiatives in the area. Providing this general overview would also provide context for the conservation practices described in lines 185-202 that were classified for the analysis.

3. The labels on figures are difficult to read. I suggest flipping the graphs so that the bars are horizontal.

Experimental design

1. The aim of the study could use clarification. Lines 98-100 state that the aim is to understand what characteristics are connected to adoption of conservation practices, but throughout the results, discussion, and conclusion the emphasis is on non-adoption. I suggest reframing the aim/introduction to reflect this.

2. Your definition of adopters and non-adopters required that both groups attended at least one outreach event, but you also interviewed respondents who did not attend outreach events and included these respondents in Figure 1 as adopters and non-adopters. Please clarify this discrepancy in the methods section.

Validity of the findings

1. The manuscript could benefit from a deeper discussion of implications of investigating non-adoption for the implementation of conservation practices. In lines 506-511, you mention implications of the results for both conservation messaging research and outreach, but the following paragraphs mainly focus on research only. I suggest that the discussion also analyze implications of results for the implementation of conservation practices. That is, how do your specific results connect to the larger outcomes discussed in lines 45-54? How could conservation practitioners utilize the results of the study (e.g. features of outreach events that non-adopters attended) to improve the efficiency or effectiveness of conservation implementation of practices introduced in outreach events? Also, how do these results compare to outcomes of other types of conservation messaging research described in lines 71-82?

2. While the justification for considering conservation practices generally on lines 213-216 is sound, future analyses could benefit from examining associations between event characteristics and adoption for specific conservation practices. For example, was there a correlation between number or type of outreach events attended and adoption of specific conservation practices? I suggest you include these potential analyses in your discussion of future research.

Additional comments

This paper addresses the characteristics of conservation outreach events associated with non-adopter agricultural land operators. The article contributes to the literature on conservation messaging and implementation and can provide important insight to both researchers and conservation practitioners. Also, the manuscript is written in professional and concise language. However, it will benefit from a clearer description of its objective and justification, specifically in regard to the relevance of examining non-adopter event features to conservation adoption and implementation.

Reviewer 2 ·

Basic reporting

Figure 3—Consider increasing the image quality here, it is difficult to read.

Lines 183-206: I might suggest creating a table out this information. I think it will improve readability---easier to scan this information in a table format than read the paragraph.

Lines 242-245: Could you provide a grounded example to illustrate this rule? I think a quick sentence or two would help to clarify this a bit.

Line 506: If the journal formatting will allow it, I would suggest breaking out your practical implications into a separate section, potentially with that title as a heading. These recommendations are important, and it would increase the readability for non-academic stakeholder if this section of the discussion was clearly demarcated in the article.

Experimental design

I’m not familiar with this approach to qualitative analysis and so I cannot offer an independent evaluation of the author’s use of QCA. That said, their explanation was clear and a reader like myself can certainly see the logic of their approach.

Validity of the findings

No comment

Additional comments

Interesting work and approaches! I found this easy to read and well prepared.

---

## Round 0.2 · accepted · Accept

· Academic Editor

Accept

Congratulations! Because you included all comments by the reviewers in the updated manuscript, I consider it as ready for publication.